# Vaccination status and its determinants among children aged 12–23 months in Tigray, northern Ethiopia: A zero-inflated Poisson regression analysis

Hailay Gebretnsae[1]*, Brhane Ayele[1], Fana Gebresilassie[1], Tsegay Hadgu[1], Hayelom Kahsay[1], Asfawosen Aregay[1], Kiros Demoz[1], Mulugeta Tilahun[1], Ataklti Gebretsadik[1], Znabu Hadush[2], Liya Mamo[2], Tsegay Wellay[2], Reda Shamie[2], Adhena Ayaliew Werkneh[2], Mebrahtu Kalayu[2], Assefa Ayallew[2], Ferehiwot Hailemariam[2], Abrham Gebrelibanos[3], Tadele Tesfean[3], Haben Haileselassie[3], Mohammedtahir Yahya[4], Melaku Abraha[5], Haftu Gebrehiwot[1], Desalegn Meresa[2], Gebrekiros Gebremichael Meles[2], Atakilti Fisseha[6], Yaynshet Gebreyohannes[6], Moges Mekonnen[6], Ashenafi Asmelash[7], Teame Zegeye[8], Mitswat Mulaw[9], Tesfu Alemu[10], Gebrehaweria Gebrekurstos[11], Rieye Esayas[11], Tsegay Berihu[11], Amanuel Haile[11], Araya Abrha Medhanyie[1,2], Mussie Alemayehu[1,2], Afework Mulugeta[2]

**1** Tigray Health Research Institute, Mekelle, Tigray, Ethiopia, **2** School of Public Health, College of Health Sciences, Mekelle University, Mekelle, Tigray, Ethiopia, **3** MARCH Research Center, College of Health Sciences, Mekelle University, Mekelle, Tigray, Ethiopia, **4** Department of Obstetrics and Gynecology, School of Medicine, College of Health Sciences, Mekelle University, Mekelle, Tigray, Ethiopia, **5** Mekelle Hamlin Fistula Center, Mekelle, Tigray, Ethiopia, **6** UNICEF-Tigray, Mekelle, Tigray, Ethiopia, **7** Mums for Mums, Mekelle, Tigray, Ethiopia, **8** Amref Health Africa, Mekele, Tigray, Ethiopia, **9** Amref Health Africa, Addis Ababa, Ethiopia, **10** UNFPA-Tigray, Mekelle, Tigray, Ethiopia, **11** Tigray Health Bureau, Mekelle, Tigray, Ethiopia

* hailish14@gmail.com

## Abstract

### Background

In Tigray, immunization services have been severely interrupted, and cases with vaccine preventable diseases become rampant due to the conflict. However, there is limited evidence on the status of child vaccination in the region. Therefore, this study aimed to assess the vaccination status and its determinants among children aged 12–23 months in Tigray, northern Ethiopia.

### Methods

A community-based cross-sectional study was conducted in August 2023 in the Tigray region, Northern Ethiopia. Using a multistage cluster sampling technique, the study included mothers of children aged 12–23 months from 19 randomly selected districts. Data were collected through a pre-tested structured questionnaire designed on Open Data Kit (ODK). Child vaccination status (the outcome variable) was measured for 14 vaccine antigens. Data analysis was conducted using R software. A zero-inflated Poisson regression model was applied to identify factors associated with child vaccination status, with statistical significance determined at a p-value of < 0.05.

**Data availability statement:** All relevant data are within the paper and its Supporting Information files.

**Funding:** The study was financially supported by UNICEF, UNFPA, Amref Health Africa, WHO and the Tigray Regional Health Bureau. Its contents are solely the responsibility of the authors and do not necessarily represent the official views of the funders. The funders had no role in study design, data collection and analysis, decision to publish, or preparation of the manuscript. There was no additional external funding received for this study.

**Competing interests:** The authors declare that they have no competing interests relevant to this manuscript.

**Abbreviations:** AIRR, Adjusted Incidence Relative Risk; AOR, Adjusted Odds Ratio; ANC, Antenatal Care; BCG, Bacillus Calmette-Guerin; CI, Confidence Interval; EPI, Expanded Programme on Immunization; HH, Household; IPV, Inactivated Polio Vaccine; ODK, Open Data Kit; OPV, Oral Polio Vaccine; PCV, Pneumococcal Conjugated Vaccine; PNC, Postnatal Care

## Results

In this study a total of 1,620 mothers of children aged 12−23 months were included. The overall proportion of fully vaccination was 59.9% (95% Confidence Interval (CI): 57.7–62.3%), while 29.4% (95% CI: 27.1–31.7%) were under-vaccinated and 10.7% (95% CI: 9.1–12.2%) were zero-dose. Being with no formal education(Adjusted Incidence Relative Risk (AIRR) = 0.94, 95% CI: 0.90–0.97), availability of routine immunization services at the nearest health facility (AIRR = 1.09, 95% CI: 1.04–1.13) and having media exposure (AIRR = 1.04, 95% CI: 1.01–1.08) were the factors associated with getting more vaccines among children. Moreover, being with no formal education (Adjusted Odds Ratio (AOR) =1.83, 95% CI: 1.21−2,75), availability of routine immunization services at the nearest health facility (AOR = 0.39, 95% CI: 0.27–0.55) and having Postnatal Care (PNC) follow-up (AOR = 0.53, 95% CI: 0.29–0.99) were the determinants for zero-dose vaccination.

## Conclusions

This study revealed that a significant proportion (40.1%) of children were left as zero-dose or under-vaccinated. To improve vaccination coverage in post-conflict settings, health policymakers should prioritize re-building immunization infrastructure and ensuring the availability of vaccination services across all levels of the healthcare system through mobilizing and allocating resources. Media campaigns should be provided to encourage mothers to vaccinate their children. Healthcare professionals should also promote timely post-natal care visits to initiate vaccinations at the appropriate age. Furthermore, it is essential to identify and reach zero-dose or under-vaccinated children through targeted catch-up vaccination efforts to ensure that no child is left unvaccinated in post war settings.

## Introduction

Globally, there were 4.9 million deaths among children under five years in 2022 [1], and Sub-Saharan Africa (SSA) accounted for 57% of this global under five deaths [2]. One-fifth (21.7%) of deaths among children under 5 years of age resulted due to vaccine preventable diseases [3]. The average mortality rate of children under five is nearly three times higher in conflict-affected countries than non-conflict affected countries [2]. Death of children under five years old can be prevented through immunization which averts 4–5 million deaths every year globally [4–6]. However, about 21 million children worldwide were zero-dose (14.5 million) or under-vaccinated (6.5 million) in 2023 [7]. Half (51.9%) and two-thirds of un-vaccinated children live in African and conflict-affected countries respectively [8,9].

The Expanded Programme on Immunization (EPI) was launched in Ethiopia in 1980 with the aim of reducing childhood morbidity and mortality [10]. Currently, Ethiopia provides routine immunization service to prevent ten vaccine-preventable diseases (measles, diphtheria, tetanus, pertussis, haemophilus influenza type B,

hepatitis B, pneumococcal disease, poliomyelitis, rotavirus infections and tuberculosis) [11–13]. According to the Ethiopian EPI national guideline, children are considered fully vaccinated when they have received one dose of Bacillus Calmette-Guerin (BCG), three doses pentavalent, three doses of polio vaccine, one dose of Inactivated Polio Vaccine (IPV), three doses of Pneumococcal Conjugate Vaccine (PCV), two doses of Rotavirus Vaccine and one dose of measles vaccine before their first birthday [11]. Although the country has shown an improvement in childhood vaccination coverage over the past few decades many children are left un-vaccinated or under-vaccinated specially in remote and conflict-affected settings [14]. The prevalence of zero-dose and under-vaccinated in conflict-affected areas of Ethiopia (Afar, Amhara, Benishangul Gumuz and Oromia regions) were 37% and 65.9%, respectively in 2022, where Tigray region was not included in that study [15]. Moreover, the prevalence of disease outbreaks is also increasing in different parts of the country [16].

Before the war which erupted on the 4th of November 2020, Tigray was one of the regions of Ethiopia with good achievement of childhood immunization coverage. The coverage of pentavalent 1, pentavalent 3, and first dose of measles and full vaccination were 95%, 83%, 83%, and 73%, respectively 2019 [14]. However, after the war, the immunization services have been seriously interrupted and the child vaccination coverage has been highly compromised [13,17,18]. Only 16% of the health facilities in the region were providing immunization services [18], and the coverage of full vaccination was declined to 20.0% in 2021 [17]. As a result, the occurrence of Vaccine Preventable Diseases (VPDs) outbreaks has increased in the region [13].

Several factors hinder the sustainable delivery of immunization services. These factors include conflict, inadequate investment in immunization programs and shortages of vaccines and supplies [6]. Conflict has been an ongoing determinant of lower immunization coverage and increased vaccine preventable disease outbreaks [19,20]. Providing childhood immunization is challenging during and post conflict because of destruction of health care systems, shortage of vaccines and supplies, displacement and security issues [19,21]. Outbreaks of vaccine preventable diseases are frequently reported in conflict- affected countries [19,20,22–24].

Several previous studies have assessed the magnitude and determinants of child vaccination status [25–30]. These studies primarily employed binary [25–28], and multinomial [29,30], logistic regression analyses, categorizing fully vaccinated children into either two or three categories without considering the number of vaccines received. Consequently, those children who are completely unvaccinated, received only one vaccine, and nearly fully vaccinated were classified all together. This method may lead to a biased conclusion in determining the factors associated with fully vaccination. In contrast, the vaccination status of fully vaccinated children should be calculated based on the total number of vaccines received, making count regression models more appropriate for identifying the determinant factors. This methodological improvement not only addresses the limitations of previous approaches but also provides more reliable and precise estimates of the factors associated with childhood immunization status. As a result, our study advances the existing literature by offering a more detailed understanding of the factors affecting child vaccination coverage and this helps to design targeted strategies to improve childhood immunization coverage, particularly in post- war settings.

In addition, following the Pretoria Cessation of Hostility Agreement (CoHA) signed on November 2, 2022 [31], several health recovery initiatives have been introduced, and health facilities have resumed providing immunization services, nonetheless the evidence regarding the vaccination status of children in the region has been limited. Therefore, this study aimed to assess child vaccination status and its determinants among children aged 12–23 months in Tigray, northern Ethiopia, using Poisson regression analysis.

## Methods

### Study design, setting and period

A community-based cross-sectional study design was conducted in six zones (western zone as a whole and some parts of southern, eastern, central and northwest zones were excluded due security reasons) of Tigray region Northern Ethiopia.

The region has seven zones which are further divided into 93 districts (57 rural and 36 urban). According to the 2007 census, the total estimated population of Tigray is approximately 7 million, and around 80% of the population lives in rural areas. Our study covered 19 randomly selected districts from 76 accessible districts of the region. Regarding the health care system, in the region there are two referral hospitals, 14 general hospitals, 24 primary hospitals, 231 health centers, and 743 health posts which were known among the well-functioning health systems in the country before the eruption of the war. However, more than 70% of health facilities were totally or partially damaged due to the war [32,33]. The study was conducted from Augus 1–30, 2023.

## Study participants

Mothers or caretakers of children aged 12–23 months were the study participants.

## Sampling procedure and sample size determination

This study was part of a large integrated health survey on utilization of maternal and child health services [34]. From the total of 13,915 households (HHs) included in the large study, 1620 households with mothers or caregivers of children aged 12–23 months were included in this specific study (Fig 1).

## Data collection procedures and quality assurances

A structured questionnaire was adapted from national immunization guidelines and other similar literature [11,12,35]. The questionnaire mainly included information on socio-demographic characteristics, reproductive health related characteristics, children's vaccination status and reasons for zero-dose or under-vaccination, and other potential information. Five days training on study objectives, how to collect the data, confidentiality and ethical concerns was provided to data collectors and supervisors. Pre-test was conducted on 5% of the sample size in three tabias that were not included in the study. Data were collected using Open Data Kit (ODK) template. Child's vaccination status was determined by self-reports from mothers/caregivers and/or immunization card reviews. In cases where the mothers/caregivers had an immunization card, the child's vaccination status was determined based on immunization card review. When an immunization card was not available, the vaccination status was assessed according to mothers' or caregivers' self-reports. Daily based data quality-check was done during the data collection period by supervisors and primary investigators.

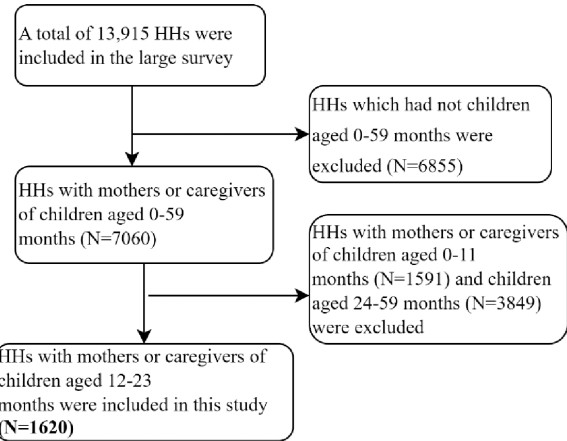

**Fig 1. Schematic presentation of participant selection procedures from the large integrated survey conducted in Tigray region, Northern Ethiopia, 2023.**

## Study variables and measurements

**Dependent variables.** The dependent variable was a count value for a total number of vaccines (Bacillus Calmette-Guerin (BCG), Oral Polio Vaccine (OPV) 1–3, Pentavalent (diphtheria, pertussis, tetanus, Hemophilus influenza type b and Hepatitis B)1–3, Pneumococcal Vaccine (PCV) 1–3, Inactivated Polio Vaccine (IPV), Rota vaccine 1 and 2 and Measles Conjugated Vaccine (MCV)1 received per child aged 12–23 months measured as 0,1, 2,3,4,5,6,7,8,9,10,11,12,13 or14.

**Fully vaccinated.** A child between 12–23 months old who received 14 doses of vaccines (one dose of Bacillus Calmette-Guerin (BCG), three doses of Oral Polio Vaccine (OPV), 3 doses of Pentavalent (diphtheria, pertussis, tetanus, Hemophilus influenza type b and Hepatitis B), 3 doses of Pneumococcal Vaccine (PCV), 1 dose of inactivated polio vaccine (IPV), 2 doses of Rota and 1 dose of measles vaccine) at the time of survey.

**Age-appropriate fully vaccinated.** A child between 12–23 months old who received all the recommended 14 doses of vaccines by his/her first birthday.

**Delayed fully vaccinated.** A child who received all recommended 14 doses of vaccines, but one or more was given later than his/her first birthday (after ≥ 12 months of age).

**Under-vaccinated.** A child between 12–23 months old who missed at least one dose of the above mentioned 14 recommended vaccines.

**Zero-dose.** A child between 12–23 months old who does not receive any dose of the above mentioned 14 recommended vaccines.

**Independent variables.** The independent variables of this study were sociodemographic variables such as: place of residence (1 = Urban and 1 = Rural), age (1 = 15–24, 2 = 25–34 and 3 = 35–49 years), marital status (1 = Currently in union and 2 = Currently not in union), occupational status (1 = Housewife, 2 = Agricultural work and 3 = Others), educational status (1 = have no formal education 2 = Primary and 3 = Secondary and above), family size(1 = < 5 and 2 = ≥ 5), sex of the child (1 = Female and 2 = Male), listening radio and/or watching television at least once a week (1 = Yes and 2 = No) and access of routine vaccination services at nearest health facility (1 = Yes and 2 = No). Reproductive and maternal health service-related variables like parity (1 = 1, 2 = 2–4 and 3= ≥ 5), ever used contraceptive (1 = Yes and 2 = No), ANC follow-up (1 = Yes and 2 = No) place of delivery (1 = Health facility and 2 = Home) and PNC follow-up (1 = Yes and 2 = No).

## Data management and analysis

The collected data were exported to SPSS version 27 for data cleaning and coding and then R statistical software Version 4.3.3 was used for analysis. Data cleaning and checking were carried out before analysis. Frequency distribution and percentages were presented using frequency tables. The dependent variable was total number of vaccines received per child measured as count 0, 1, 2,3,4,5,6,7,8,9,10,11,12,13 or 14. Since our dependent variable was count data, we used Poisson regression analysis to identify the determinant factors. Poisson regression is the most popular model for count dependent variable [36–38]. To select the best model of Poisson regressions, Akaike's Information Criteria (AIC), Bayesian's Information Criteria (BIC) and Log-Likelihood (LL) were compared and the model with lowest value of AIC) and BIC and with the greatest LL was selected as best model [39–41]. Accordingly, the Zero-Inflated Poisson (ZIP) had the smallest values for AIC and BIC and with greatest value for LL, which makes it best model to fit the data on the number of vaccines received per child. The strength of association was assessed using Adjusted Incidence Relative Risk (AIRR) with their respective 95% confidence interval for count part and Adjusted Odds Ratio (AOR) with their respective 95% confidence interval for zero part. Finally, the variables with p-value less than 0.05 were identified as determinant factors.

## Ethical consideration

An ethical clearance for the large survey was received from Tigray Health Research Institute Institutional Review Board (reference number: THRI/4031/1099/15). Additionally, an official support letter was secured from Tigray Health Bureau.

After explaining the study's purpose, risks, benefits and their right to participate or not, verbal informed consent was obtained from participants whose age was 18 and above years before the interview. Furthermore, if the participants' age was below 18 years, verbal informed assent and consent were obtained from themselves and their parents/guardians respectively. Data were collected anonymously, and confidentiality was maintained. Those zero-dose or under- vaccinated children were vaccinated according to the national EPI guideline.

## Results

### Socio-demographic characteristics of mothers and children

A total of 1620 mothers or caregivers of children aged 12–23 months were included in the study. The mean (± Standard Deviation (SD)) age of the respondents was 30.0 (± 7.2) years with nearly half (48.5%, n = 785) of them were in the age group of 25–34 years. Above three-fourths (79.9%, n = 1295) lived in rural areas, 806 (49.8%) had no formal education, and 917(56.6%) were housewives. About 938(57.9%) HHs of respondents had at least five HH members and one-fifth (19.9%, n = 322) of respondents listened to radio or watched television at least once per week (Table 1).

### Reproductive and maternal health service utilization characteristics of mothers

One-fifth (20.0%, n = 324) of mothers were para one, and 86(5.3%) of them ever had history of stillbirth in their lifetime. Regarding maternal health service utilization, 40.7%(n = 659) had antenatal care follow-up, 30.4%(n = 492) delivered at health facilities, and 15.8%(n = 256) had at least one postnatal care visit for their index child (Table 2).

### Child vaccination status

The data on child vaccination coverage was collected based on child immunization card 865(53.4%) and mothers' or care-givers' recall 755(46.6%). One-tenth (10.73%, n = 173) of children were zero-dose and six out of every ten (n = 971, 59.9%) children had received all the recommended vaccines. The number of vaccinations showed that the variance (22.72) is greater than the mean (11.08) indicating over-dispersion (Table 3). The proportion of fully vaccinated (59.9%: 95%CI: 57.7–62.3%) was composed of 31.4% (95%CI: 29.0–33.6%) of children with age-appropriate fully vaccinated and 28.6% (95%CI: 26.5–30.9%) with delayed fully vaccinated (Fig 2).

### Reasons for zero- dose or under-vaccination

In cases of children with zero-dose or under-vaccination, major reasons reported were: lack of vaccine (47.0%), security concern (12.3%), closure of health facility (11.0%) and vial was not opened for small number (8.7%) (Fig 3).

### Model selection criteria

The mean of vaccines received was less than the variance; this suggests that the Poisson regression may not be suitable. Nonetheless, further statistical tests were conducted to determine the optimal model. The AIC, BIC and Log-Likelihood (LL) were used as further tests of the goodness of fit for model comparison and selection. Although the ZIP and ZINB had similar LL, the ZIP had the smallest values for AIC and BIC, which makes it best model to fit the data on the number of vaccines received per child (Table 4).

### Factors associated with child vaccination status

Maternal education and availability of routine immunization services at the nearest health facility were significantly associated with getting more vaccines and zero-dose. Additionally, listening to radio or watching television at least once a week was associated with getting more vaccines while PNC follow-up had a significant negative association with zero-dose status among children aged 12–23 months.

**Table 1. Socio-demographic characteristics of mother and children aged 12- 23 months in Tigray Northern Ethiopia 2023 (n = 1620).**

| Variables | Frequency | Percent |
|---|---|---|
| **Maternal age (in years)** | | |
| 15–24 | 368 | 22.7 |
| 25–34 | 785 | 48.5 |
| 35–49 | 467 | 28.8 |
| **Place of residence** | | |
| Urban | 325 | 20.1 |
| Rural | 1295 | 79.9 |
| **Participant type** | | |
| Resident | 1448 | 89.4 |
| Internal Displaced People (IDP) | 172 | 10.6 |
| **Maternal religion** | | |
| Orthodox | 1586 | 97.9 |
| Muslim | 34 | 2.1 |
| **Maternal marital status** | | |
| Currently not in union* | 166 | 10.2 |
| Currently in union | 1454 | 89.8 |
| **Maternal educational status** | | |
| Have no formal education | 806 | 49.8 |
| Primary | 427 | 26.4 |
| Secondary and above | 287 | 23.9 |
| **Maternal occupation** | | |
| Housewife | 917 | 56.6 |
| Agricultural work | 543 | 33.5 |
| Others** | 160 | 9.9 |
| **Sex of index child** | | |
| Male | 806 | 49.8 |
| Female | 814 | 50.2 |
| **Family size** | | |
| ≤5 | 938 | 57.9 |
| >5 | 682 | 42.1 |
| **Numbers of under five children in the household** | | |
| 1 | 975 | 60.2 |
| 2 | 621 | 38.3 |
| 3 | 24 | 1.5 |
| **Listening to the radio and/or watching television at least once a week** | | |
| Yes | 322 | 19.9 |
| No | 1298 | 80.1 |
| **Availability of routine immunization services at nearest health facility** | | |
| Yes | 1271 | 78.5 |
| No | 349 | 21.5 |

* Currently not in union (single, widowed, divorced and separated),

** Others (governmental employed, self-employed and merchant).

**Table 2. Reproductive and maternal health service utilization characteristics of mothers in Tigray, Northern Ethiopia 2023 (n = 1620).**

| Variables | Frequency | Percent |
|---|---|---|
| **Parity** | | |
| 1 | 324 | 20.0 |
| 2-4 | 821 | 50.7 |
| ≥5 | 475 | 29.3 |
| **Ever had history of stillbirth** | | |
| Yes | 86 | 5.3 |
| No | 1534 | 94.7 |
| **Ever had history of abortion** | | |
| Yes | 290 | 17.9 |
| No | 1330 | 82.1 |
| **Ever used contraceptive methods** | | |
| Yes | 962 | 59.4 |
| No | 658 | 40.6 |
| **ANC* follow-up** | | |
| Yes | 659 | 40.7 |
| No | 961 | 59.3 |
| **Place of delivery** | | |
| Health facility | 492 | 30.4 |
| Home | 1128 | 69.6 |
| **PNC** follow-up** | | |
| Yes | 256 | 15.8 |
| No | 1364 | 84.2 |

*ANC-Antenatal care, **PNC-Postnatal care.

**Table 3. Number of vaccines received per child in Tigray, Northern Ethiopia 2023 (n = 1620).**

| Number of received vaccines | Frequency | Percent |
|---|---|---|
| 0 | 173 | 10.7 |
| 1 | 10 | 0.6 |
| 2 | 5 | 0.3 |
| 3 | 5 | 0.3 |
| 4 | 39 | 2.4 |
| 5 | 47 | 2.9 |
| 6 | 23 | 1.4 |
| 7 | 7 | 0.4 |
| 8 | 23 | 1.4 |
| 9 | 74 | 4.6 |
| 10 | 57 | 3.5 |
| 11 | 34 | 2.1 |
| 12 | 21 | 1.3 |
| 13 | 131 | 8.1 |
| 14 | 971 | 59.9 |
| Total | 1620 | 100.0 |
| Mean | 11.08 | |
| Variance | 22.72 | |

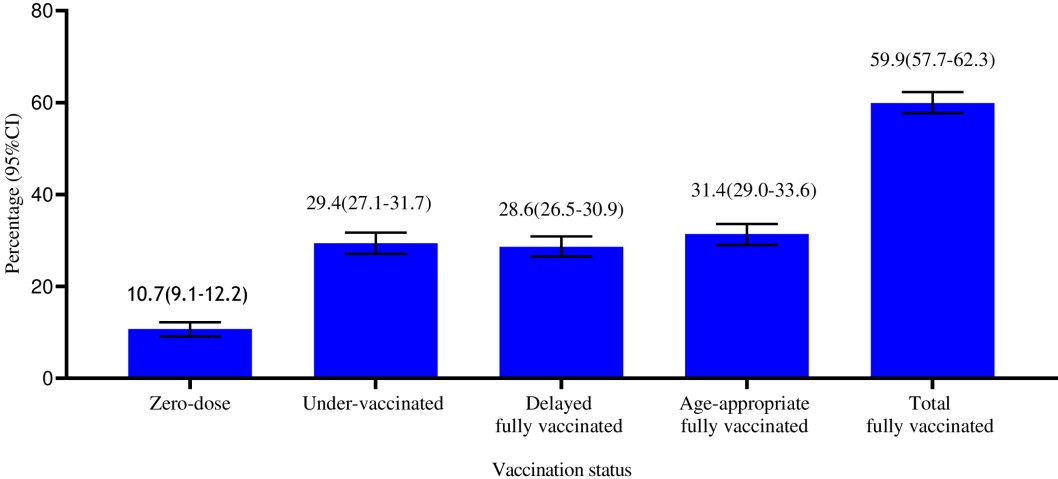

**Fig 2. Vaccination status among children aged 12-23 months in Tigray, Northern Ethiopia 2023 (n = 1620).**

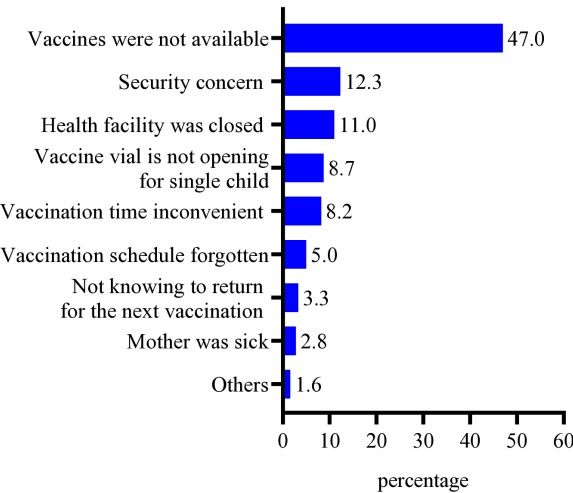

**Fig 3. Reasons for zero- dose or under-vaccination among children aged 12-23 months in Tigray, Northern Ethiopia 2023 (n = 649).**

**Table 4. Model selection criteria.**

| Model | AIC | BIC | LL |
|---|---|---|---|
| Poisson | 11375.511 | 11456.36 | −5672.8 |
| NB | 10550.356 | 10636.6 | −5259.2 |
| ZIP | 8594.591 | 8756.296 | −4267 |
| ZINB | 8596.591 | 8763.687 | −4267 |

NB: Negative Binomial, ZIP: Zero-Inflated Poisson, ZINB: zero-Inflated Negative Binomial.

Children born from mothers who had no formal education were 6% less likely (AIRR = 0.94, 95% CI: 0.90–0.97) to get more vaccines than those born to educated mothers. Children who had availability of routine immunization services at the nearest health facility were 9% more likely (AIRR = 1.09, 95% CI: 1.04–1.13) to get more vaccines when compared to children who had not availability of routine immunization services. Media exposure was significantly associated with getting more vaccines; children born from mothers who listened radio and/or watched television at least once a week were 4% more likely (AIRR = 1.04, 95% CI: 1.01–1.08) to get more vaccines when compared with those children born from mothers who did not listen radio or watch television at least once a week.

Furthermore, children born to mothers who had no formal education were 1.83 times higher odds of being zero-dose (AOR = 1.83, 95% CI: 1.21–2.75) when compared to those born to educated mothers. Children who had availability of routine immunization services at the nearest health facility were 61% less likely (AOR = 0.39, 95% CI: 0.27–0.55) to be zero-dose when compared to children who had no availability of routine immunization services at nearby health facility. Lastly, children born to mothers who had PNC follow-up were 47% less likely (AOR = 0.53, 95% CI: 0.29–0.99) to be zero-dose when compared to those born to mothers who had not received PNC follow-up (Table 5).

## Discussion

This study assessed the vaccination status and its determinants among children aged 12–23 months in Tigray, northern Ethiopia. The results indicated that 59.9% of the children were fully vaccinated, while 29.4% were under-vaccinated and 10.7% were zero-dose. Maternal educational level and availability of routine immunization services at the nearest health facility were significantly associated with getting more vaccines and zero-dose. Additionally, listening radio or watching television at least once per week was associated with getting more vaccines, and postnatal care (PNC) follow-up was associated with zero-dose status among children aged 12–23 months.

Timely vaccination is important to minimize individual vulnerability and prevent disease outbreaks within communities [42–44]. Our study shows that only one-third (34.7%) of children had received fully vaccination before first year of life. This is almost in line with that reported from Gambia which revealed that 36.7% of children had received all their recommended vaccines on-time [45]. However, our finding was lower when compared with the results of other studies conducted in Ethiopia (55.9%) [46], Uganda (45.6%) [47], Sierraleone (56%) [48], and Cameroon (73,4%) [49] of children were completed all the recommended vaccinations before one year of age. Furthermore, the total fully vaccinated had declined from 73% prewar [14] to 59.9%. Declining in child vaccination coverages were reported from other findings of conflict affected countries [19,21,24,50–52]. Our finding in the total fully vaccinated is almost the same with that reported from Ethiopia (57.4%) [53]. However, our result was lower than other research findings in Ethiopia ranged from 66.4%− 82.3% [26,46,54–58], Gambia (66%) [59], Senegal (72.2%) [60], Kenya (76.6%) [61], Cameroon (85.9%) [49], and Bangladesh (86%) [62]. This could be due to the disruption of Tigray's health system, including the immunization service following the conflict. This study indicated that a large proportion (40.1%) of children remained un-vaccinated or under-vaccinated. This could increase the risk of VPDs and can have a negative influence on herd immunity of the community.

In this study, maternal education was identified as a determinant factor of childhood vaccination uptake. This finding is supported by previous studies which revealed that maternal education was predictor variable of fully vaccination [58,59,63–67]. Educated mothers have more understanding of the importance of child vaccination than mothers with no formal education. Educational level has positive impact on healthcare seeking of the mothers by empowering them on decision making to get healthcare services including vaccination [60].

In the current study, children who had availability of routine immunization services at their nearest health facility had higher chance to initiate their vaccinations and to get more vaccines than children who had no availability of routine immunization services at their nearby health facility. This could be due to most of the health facilities in Tigray being totally or partially damaged following the conflict [32,33], which affected the availability of routine immunization services [68,69]. Furthermore, our findings indicated that lack of vaccines (47.0%), security concerns (12.3%) and closure of health facility

**Table 5. Zero-inflated poison regression for vaccination status among children aged 12- 23 months in Tigray, Northern Ethiopia, 2023 (n = 1620).**

| Variables | Count part | Zero part |
|---|---|---|
| | AIRR (95%CI) | AOR (95%CI) |
| **Place of residence** | | |
| Urban | 1.03(0.99-1.07) | 1.49(0.87-2.55) |
| Rural | 1 | 1 |
| **Maternal marital status** | | |
| Currently in union | 1.01(0.96-1.06) | 0.60(0.31-1.15) |
| Currently not in union* | 1 | 1 |
| **Maternal educational status** | | |
| Illiterate | **0.94(0.90-0.97) *** | **1.83(1.21-2.75) *** |
| literate | 1 | 1 |
| **Maternal occupation** | | |
| Housewife | 1.01(0.96-1.06) | 1.45(0.71-2.97) |
| Agricultural work | 0.96(0.91-1.02) | 1.54(0.71-3.33) |
| Others** | 1 | 1 |
| **Family size** | | |
| ≤5 | 1.00(0.96-1.04) | 0.87(0.56-1.36) |
| >5 | 1 | 1 |
| **Listening radio and/or watching television at least once a week** | | |
| Yes | **1.04(1.01-1.08) *** | 1.23(0.78-1.94) |
| No | 1 | 1 |
| **Availability of routine immunization service at nearest health facility** | | |
| Yes | **1.09(1.04-1.13) *** | **0.39(0.27-0.55) *** |
| No | 1 | 1 |
| **Parity** | | |
| 1 | 0.98(0.92-1.03) | 1.25(0.66-2.36) |
| 2-4 | 0.98(0.94-1.03) | 1.11(0.71-1.75) |
| ≥ 5 | 1 | 1 |
| **Ever used contraceptive methods** | | |
| Yes | 1.00(0.96-1.03) | 0.77(0.55-1.09) |
| No | 1 | 1 |
| **ANC follow-up** | | |
| Yes | 1.00(0.97-1.04) | 1.00(0.67-1.48) |
| No | 1 | 1 |
| **Place of delivery** | | |
| Health facility | 1.01(0.97-1.05) | 0.89(0.56-1.42) |
| Home | 1 | 1 |
| **PNC follow-up** | | |
| Yes | 1.02(0.98-1.07) | **0.53(0.29-0.99) *** |
| No | 1 | 1 |

*Statistically significant at 0.05 < p < 0.01.

**Statistically significant at 0.01 < p < 0.001.

***Statistically significant at p < 0.001.

(11.0%) were major reasons for zero-dose or under-vaccination. This finding is consistent with previous studies which reported that interruptions in vaccine supplies, inaccessibility of health facilities, power interruption and refrigerator break-down were the main reasons for the high dropout rate and low coverage of childhood vaccination in conflict-affected areas of Ethiopia [15,54].

Having access to health information through media could play a pivotal role in improving mothers' awareness regarding child immunization [61,70]. In the current study, media exposure was significantly associated with child vaccination uptake. The probability of getting more vaccines among children born from mothers who listened to radio and/or watched TV at least once a week were more likely to get more vaccines compared to children born from mothers who did not listen radio and/or watch TV at least once a week. This finding is consistent with the study reported that exposure to media was significant predictor of full vaccination [64,65,71]. This may be explained by the effectiveness of media in information dissemination. In addition, media could facilitate behavioral changes allowing for the adoption of different behaviors including child vaccination uptake [71].

In this study, the odds of zero-dose were higher among children whose mother had no postnatal care. This finding is in line with previous studies which revealed that children born from mothers who had no PNC follow-up were at high risk to become zero-dose as compared children born from mothers who had PNC follow-up [72]. Similarly, previous studies reported that PNC was significant determinant of full vaccination [64–66,73–76]. Vaccination is an essential component of PNC and PNC visits offer an opportunity to mothers to learn about vaccination. If the mothers receive PNC, at the same time they may start vaccination to their children or they could receive enough information from healthcare providers about benefits of immunization and when they start vaccination of their children [77].

## Strengths and limitations of the study

The current study is the first to investigate the vaccination status and its determinants among children aged 12–23 months in the study area after the war, so that it is critically important to the health program managers and partners in restoring immunization services and improving child immunization coverage. However, in case of child immunization card was not available, caregivers' recall was used to evaluate whether the child had been vaccinated or not, this may lead to recall bias. Moreover, some districts were not included in this study due to security reasons and vaccination coverage might be lower than the coverage estimated in this study.

## Conclusions

This study revealed that a significant proportion (40.1%) of children were left as zero-dose or under-vaccinated. To improve vaccination coverage in post-conflict settings, health policymakers should prioritize re-building immunization infrastructure and ensuring the availability of vaccination services across all levels of the healthcare system through mobilizing and allocating resources. Media campaigns should be provided to encourage mothers to vaccinate their children, emphasizing the importance of timely initiation of immunization and its long-term benefits. Healthcare professionals should also promote timely post-natal care visits to initiate vaccinations at the appropriate age. Furthermore, it is essential to identify and reach zero-dose or under-vaccinated children through targeted catch-up vaccination efforts to ensure that no child is left unvaccinated in post war settings.

## Supporting information

**S1 Checklist. This S1 Checklist contains the STROBE check list of the manuscript.**
(DOCX)

**S1 Data. Dataset of the study.**
(SAV)

## Acknowledgments

We would like to acknowledge to the selected District Health Offices, Mums for Mums (MfM) and Hamlin fistula center Mekelle branch for their technical support during the data collection. We also thank the supervisors, data collectors and study participants for their contributions to the accomplishment of this research.

## Author contributions

**Conceptualization:** Hailay Gebretnsae, Brhane Ayele, Tsegay Hadgu, Hayelom Kahsay, Znabu Hadush, Araya Abrha Medhanyie, Mussie Alemayehu, Afework Mulugeta.

**Data curation:** Hailay Gebretnsae, Fana Gebresilassie, Liya Mamo.

**Formal analysis:** Hailay Gebretnsae, Fana Gebresilassie.

**Funding acquisition:** Hailay Gebretnsae, Brhane Ayele, Tsegay Hadgu, Hayelom Kahsay, Znabu Hadush, Atakilti Fisseha, Yaynshet Gebreyohannes, Moges Mekonnen, Gebrehaweria Gebrekurstos, Rieye Esayas, Tsegay Berihu, Amanuel Haile, Araya Abrha Medhanyie, Mussie Alemayehu.

**Investigation:** Hailay Gebretnsae, Brhane Ayele, Tsegay Hadgu, Hayelom Kahsay, Znabu Hadush, Araya Abrha Medhanyie, Mussie Alemayehu, Afework Mulugeta.

**Methodology:** Hailay Gebretnsae, Brhane Ayele, Fana Gebresilassie, Mussie Alemayehu.

**Project administration:** Hailay Gebretnsae, Brhane Ayele, Tsegay Hadgu, Hayelom Kahsay, Znabu Hadush, Araya Abrha Medhanyie, Mussie Alemayehu, Afework Mulugeta.

**Resources:** Hailay Gebretnsae, Brhane Ayele, Tsegay Hadgu, Hayelom Kahsay, Znabu Hadush, Atakilti Fisseha, Yaynshet Gebreyohannes, Moges Mekonnen, Ashenafi Asmelash, Teame Zegeye, Mitswat Mulaw, Tesfu Alemu, Gebrehaweria Gebrekurstos, Rieye Esayas, Tsegay Berihu, Amanuel Haile, Araya Abrha Medhanyie, Mussie Alemayehu, Afework Mulugeta.

**Software:** Hailay Gebretnsae, Liya Mamo, Haftu Gebrehiwot, Desalegn Meresa, Gebrekiros Gebremichael Meles.

**Supervision:** Hailay Gebretnsae, Brhane Ayele, Fana Gebresilassie, Tsegay Hadgu, Hayelom Kahsay, Asfawosen Aregay, Kiros Demoz, Mulugeta Tilahun, Ataklti Gebretsadik, Znabu Hadush, Liya Mamo, Tsegay Wellay, Reda Shamie, Adhena Ayaliew Werkneh, Mebrahtu Kalayu, Assefa Ayallew, Ferehiwot Hailemariam, Abrham Gebrelibanos, Tadele Tesfean, Haben Haileselassie, Mohammedtahir Yahya, Melaku Abraha, Gebrehaweria Gebrekurstos, Amanuel Haile, Araya Abrha Medhanyie, Mussie Alemayehu, Afework Mulugeta.

**Validation:** Hailay Gebretnsae, Brhane Ayele, Fana Gebresilassie, Tsegay Hadgu, Hayelom Kahsay, Asfawosen Aregay, Kiros Demoz, Znabu Hadush, Liya Mamo, Mohammedtahir Yahya, Melaku Abraha, Atakilti Fisseha, Moges Mekonnen, Gebrehaweria Gebrekurstos, Araya Abrha Medhanyie, Mussie Alemayehu, Afework Mulugeta.

**Visualization:** Hailay Gebretnsae, Fana Gebresilassie, Mulugeta Tilahun, Ataklti Gebretsadik, Znabu Hadush, Liya Mamo.

**Writing – original draft:** Hailay Gebretnsae.

**Writing – review & editing:** Hailay Gebretnsae, Brhane Ayele, Fana Gebresilassie, Tsegay Hadgu, Hayelom Kahsay, Asfawosen Aregay, Kiros Demoz, Mulugeta Tilahun, Ataklti Gebretsadik, Znabu Hadush, Liya Mamo, Tsegay Wellay, Reda Shamie, Adhena Ayaliew Werkneh, Mebrahtu Kalayu, Assefa Ayallew, Ferehiwot Hailemariam, Abrham Gebrelibanos, Tadele Tesfean, Haben Haileselassie, Mohammedtahir Yahya, Melaku Abraha, Haftu Gebrehiwot, Desalegn Meresa, Gebrekiros Gebremichael Meles, Yaynshet Gebreyohannes, Moges Mekonnen, Ashenafi Asmelash, Teame Zegeye, Mitswat Mulaw, Tesfu Alemu, Gebrehaweria Gebrekurstos, Rieye Esayas, Araya Abrha Medhanyie, Mussie Alemayehu, Afework Mulugeta.

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
