## [Decision Letter · Decision Letter 0]

Dear Dr. Gebretnsae,

Thank you for submitting your manuscript to PLOS ONE. After careful consideration, we feel that it has merit but does not fully meet PLOS ONE’s publication criteria as it currently stands. Therefore, we invite you to submit a revised version of the manuscript that addresses the points raised during the review process.

**ACADEMIC EDITOR: **

We look forward to receiving your revised manuscript.

Kind regards,

Omar Enzo Santangelo

Academic Editor

PLOS ONE

Journal Requirements:

2. In the ethics statement in the Methods, you have specified that verbal consent was obtained. Please provide additional details regarding how this consent was documented and witnessed, and state whether this was approved by the IRB

3. You indicated that you had ethical approval for your study. In your Methods section, please ensure you have also stated whether you obtained consent from parents or guardians of the minors included in the study or whether the research ethics committee or IRB specifically waived the need for their consent.

“The study was financially supported by UNICEF, UNFPA, Amref Health Africa and Tigray Reginal Health Bureau. Its contents are solely the responsibility of the authors and do not necessarily represent the official views of the funders. The funders had no role in study design, data collection and analysis, decision to publish or preparation of the manuscript.”

Reviewers' comments:

Reviewer's Responses to Questions

**Comments to the Author**

1. Is the manuscript technically sound, and do the data support the conclusions?

Reviewer #1: Partly

Reviewer #2: Yes

2. Has the statistical analysis been performed appropriately and rigorously?

Reviewer #1: Yes

Reviewer #2: Yes

3. Have the authors made all data underlying the findings in their manuscript fully available?

Reviewer #1: Yes

Reviewer #2: Yes

4. Is the manuscript presented in an intelligible fashion and written in standard English?

Reviewer #1: Yes

Reviewer #2: Yes

Reviewer #1: The study addresses a highly relevant and timely topic in the context of post-conflict regions like Tigray, Ethiopia. It provides valuable insights into child vaccination coverage after a prolonged conflict that disrupted health services. However, similar studies have been conducted in other conflict-affected areas. The novel aspect of this work is its focus on the use of zero-inflated Poisson regression to assess the determinants of vaccination, which allows for a more nuanced understanding of both zero-dose and under-vaccinated populations.

1. Could you elaborate on how this study contributes to the existing literature beyond the use of zero-inflated Poisson regression? What new insights does this methodology provide that were not possible with previous approaches?

2. How generalizable are the findings to other conflict-affected regions beyond Tigray?

Methodology: The study follows a cross-sectional design, which is appropriate for assessing vaccination status at a specific time point. The use of multistage cluster sampling is a strength, ensuring that the sample is representative of the region. The decision to use the zero-inflated Poisson regression model is justified, as it accounts for the overdispersion in the data. The sample size (1,620 mothers) is robust, and the data collection methods (both immunization cards and caregiver recall) are appropriate, although caregiver recall may introduce some bias.

1. What steps were taken to minimize recall bias, especially given the reliance on caregiver memory for vaccination status?

2. Could you explain why certain districts were excluded from the study and how this might impact the overall generalizability of your findings?

Results: The results are clearly presented, with detailed statistical analysis. The findings that maternal education, media exposure, and availability of routine immunization services are significant determinants of vaccination status are in line with expectations. However, the study's conclusion that 40.1% of children were either zero-dose or under-vaccinated is concerning and underscores the need for targeted interventions.

1. Could you provide more detailed recommendations on how health policymakers can improve vaccination rates in this post-conflict context?

2. Were there any unexpected findings in the analysis that merit further discussion?

Discussion:

The discussion links the results to previous research and highlights the importance of restoring health services in conflict-affected regions. However, it lacks depth in addressing some of the limitations (e.g., the use of recall data and the exclusion of some districts for security reasons).

1. Could the authors discuss how selection bias (due to district exclusion) might affect generalizability?

2. Have the authors considered how other cultural or systemic barriers (e.g., trust in health services) may have played a role in vaccination outcomes?

Conclusions:

The conclusion is consistent with the results and addresses the importance of media campaigns and health service restoration. However, the call for catch-up vaccination programs could be expanded with more specific recommendations for policy or practice.

1. Can the authors suggest concrete policy recommendations based on the findings, especially in terms of mobilizing resources for vaccination campaigns?

Reviewer #2: This study describes immunization uptake in 12-23-month-old children in Tigray. Roughly 60% of children were found to be fully vaccinated according to the WHO/GAVI recommendation of 14 total vaccines, 30% were not fully vaccinated and 10% had received no vaccinations. The study also explored the reasons for children not having received any (or all) of the required vaccinations. One limitation of the study is that only about half of the subjects could provide a written record of their immunization status, and so the others were assessed based on oral reports from their caregivers. Purely relying on the written record would not be desirable since it may bias against detection of unvaccinated children, as they would be more likely to have no written record, causing them to be removed for the analysis. However it may be a useful check on the accuracy of the conclusions to compare the number of fully and partially vaccinated children reported using only the subset of those with written records with those rates in the overall study population. This may help to strengthen confidence in the accuracy of the overall conclusions.

**Do you want your identity to be public for this peer review?** For information about this choice, including consent withdrawal, please see our Privacy Policy

Reviewer #1: No

Reviewer #2: No

---

## [Author Response · Author response to Decision Letter 1]

6 May 2025

Responses to editor’s and reviewers’ comments

Journal Requirements:

Editor ’s comment: 1. Please ensure that your manuscript meets PLOS ONE's style requirements, including those for file naming.

Response to editor’s comment: Thank you for your guidance. We have carefully revised the manuscript to ensure they meet PLOS ONE’s style requirements, including the file naming.

Editor ’s comment: 2. In the ethics statement in the Methods, you have specified that verbal consent was obtained. Please provide additional details regarding how this consent was documented and witnessed, and state whether this was approved by the IRB.

Response to editor’s comment: Thank you for your comment. The study protocol, including the verbal consent procedure, was approved by the Institutional Ethical Review Board of Tigray Health Research Institute (reference number: THRI/4031/1099/15). The information sheet and consent form were incorporated into the data collection tool, and data collectors documented the verbally informed consent by signing the respective section before conducting each interview. Informed Verbal consent was used because many of the study participants were expected to have limited literacy. Additionally, the study involved a survey with minimal risk to participants, further supporting the use of verbal consent.

Editor ’s comment: 3. You indicated that you had ethical approval for your study. In your Methods section, please ensure you have also stated whether you obtained consent from parents or guardians of the minors included in the study or whether the research ethics committee or IRB specifically waived the need for their consent.

Response to editor’s comment: Thank you for your comment. We confirm that verbal informed consent was obtained from the parents or guardians of all minor participants included in the study. This has now been clearly stated in the Methods section in the sub section of Ethical consideration on page 10, and line221-223 now reads,” After explaining the study’s purpose, risks, benefits and their right to participate or not, verbal informed consent was obtained from each study participants before the interview. Additionally, if the participants were below the age of 18 years written informed assent was obtained from their parents/guardians”. The study protocol, including the process for obtaining parental or guardian consent, was approved by the Institutional Review Board.

Editor ’s comment: 4. Thank you for stating in your Funding Statement:

“The study was financially supported by UNICEF, UNFPA, Amref Health Africa and Tigray Reginal Health Bureau. Its contents are solely the responsibility of the authors and do not necessarily represent the official views of the funders. The funders had no role in study design, data collection and analysis, decision to publish or preparation of the manuscript.”

Response to editor’s comment: Thank you for your comment and we have revised as you suggested. The study was financially supported by UNICEF, UNFPA, Amref Health Africa, WHO and the Tigray Regional Health Bureau. Its contents are solely the responsibility of the authors and do not necessarily represent the official views of the funders. The funders had no role in study design, data collection and analysis, decision to publish, or preparation of the manuscript. There was no additional external funding received for this study.

Editor ’s comment :5. When completing the data availability statement of the submission form, you indicated that you will make your data available on acceptance. We strongly recommend all authors decide on a data sharing plan before acceptance, as the process can be lengthy and hold up publication timelines. Please note that, though access restrictions are acceptable now, your entire data will need to be made freely accessible if your manuscript is accepted for publication. This policy applies to all data except where public deposition would breach compliance with the protocol approved by your research ethics board. If you are unable to adhere to our open data policy, please kindly revise your statement to explain your reasoning and we will seek the editor's input on an exemption. Please be assured that, once you have provided your new statement, the assessment of your exemption will not hold up the peer review process.

Response to editor’s comment: Thank you for your feedback. We confirm that the dataset will be uploaded to the journal for open access upon acceptance of the manuscript, if required.

Reviewer #1:

Introduction:

Reviewer’s comment: 1. Could you elaborate on how this study contributes to the existing literature beyond the use of zero-inflated Poisson regression? What new insights does this methodology provide that were not possible with previous approaches?

Response to review’s comment: Thank you for your insightful comment. We have elaborated the contribution of our methodology over existing literature in the revised introduction section. The revised introduction section on page 5 and 6, and line 122-127now reads,” … This methodological improvement not only addresses the limitations of previous approaches but also provides more reliable and precise estimates of the factors associated with childhood immunization status. As a result, our study advances the existing literatures by offering a more detailed understanding of the factors affecting child vaccination coverage and this helps to design targeted strategies to improve childhood immunization coverage, particularly in post- war settings”.

Reviewer’s comment: 2. How generalizable are the findings to other conflict-affected regions beyond Tigray?

Response to review’s comment: Thank you for your insightful comment. We have incorporated your suggestion in the revised introduction section. The revised introduction section on 6, and line 126 and 127 now reads,” … and this helps to design targeted strategies to improve childhood immunization coverage, particularly in post- war settings”.

Methodology:

Reviewer’s comment: 1. What steps were taken to minimize recall bias, especially given the reliance on caregiver memory for vaccination status?

Response to review’s comment: Thank you for your insightful comment. To minimize recall bias, we employed several strategies and ensured that interviewers were extensively trained to probe gently and consistently without leading respondents. In cases where immunization cards were unavailable, interviewers followed a structured approach to assist caregivers in recalling their child’s vaccination status. They referenced specific vaccination timelines (e.g., at birth, 6 weeks, 10 weeks, 14 weeks, 9 months, and 15 months) and inquired about the sequence of vaccinations (e.g., first, second, and third doses, as well as vaccines given at 9 months and 15 months). Interviewers also asked about the types of vaccines administered, such as whether the child received oral vaccines, and for injectable vaccines, they inquired about the number of doses and the sites of administration (e.g., both thighs and arms). This approach, along with related techniques, helped to ensure that the recall was as accurate and comprehensive as possible despite the absence of immunization cards. Such methods are recommended by the World Health Organization (WHO) Immunization Coverage Monitoring guideline for assessing childhood vaccination status when immunization cards are not available.

Reviewer’s comment: 2. Could you explain why certain districts were excluded from the study and how this might impact the overall generalizability of your findings?

Response to review’s comment: Thank you for your comment. Out of the 93 districts in Tigray, 17 were excluded from the study due to security concerns. These districts are not under the governance of the Tigray interim administration, as they are controlled by external forces. As a result, they were not included in our study. However, we have addressed the issue of generalizability in the limitations section. The revised "Strengths and Limitations" section, found on page22, lines 371and 372, now reads “…Moreover, some districts were not included in this study due to security reasons and vaccination coverage might be lower than the coverage estimated in this study”.

Results:

Reviewer’s comment: 1. Could you provide more detailed recommendations on how health policymakers can improve vaccination rates in this post-conflict context?

Response to review’s comment: Thank you for your valuable comment. We have incorporated recommendations on how health policymakers can improve childhood vaccination rates in post-conflict settings in the revised conclusion section. The revised conclusions section, found on page 22, and lines 375-378, now reads as follows” To improve vaccination coverage in post-conflict settings, health policymakers should prioritize re-building immunization infrastructure and ensuring the availability of vaccination services across all levels of the healthcare system through mobilizing and allocating resources.”

Reviewer’s comment: 2. Were there any unexpected findings in the analysis that merit further discussion?

Response to review’s comment: Thank you for your thoughtful question. In Tigray, more than 70% of health facilities were either completely or partially destroyed due to the conflict, leading to severe disruptions in routine childhood vaccination services. Therefore, the high proportion of zero-dose and under-vaccinated children observed in our study was anticipated. Our findings did not reveal any major unexpected outcomes; rather, they aligned with our initial assumptions regarding the impact of healthcare system disruption.

Discussion:

Reviewer’s comment: 1. Could the authors discuss how selection bias (due to district exclusion) might affect generalizability?

Response to review’s comment: Thank you for your insightful comment. We have addressed the issue of generalizability in the limitations section. The revised "Strengths and Limitations" section, found on page 22 and lines 371 and 372, now reads as follows, “Moreover, some districts were not included in this study due to security reasons and vaccination coverage might be lower than the coverage estimated in this study”.

Reviewer’s comment: 2. Have the authors considered how other cultural or systemic barriers (e.g., trust in health services) may have played a role in vaccination outcomes?

Response to review’s comment: Thank you for your comment. Cultural barriers were not specifically addressed in our study, as our focus was primarily on socio-demographic and economic factors, reproductive and maternal health service utilization, healthcare access, and other key predictors relevant in the post-conflict context. These variables were prioritized based on their expected influence on vaccination coverage during this period.

Conclusions:

Reviewer’s comment: 1. Can the authors suggest concrete policy recommendations based on the findings, especially in terms of mobilizing resources for vaccination campaigns?

Response to review’s comment: Thank you for your comment. We have made the suggested revisions as per your recommendation.

Reviewer #2:

Reviewer’s comment: One limitation of the study is that only about half of the subjects could provide a written record of their immunization status, and so the others were assessed based on oral reports from their caregivers. Purely relying on the written record would not be desirable since it may bias against detection of unvaccinated children, as they would be more likely to have no written record, causing them to be removed for the analysis. However, it may be a useful check on the accuracy of the conclusions to compare the number of fully and partially vaccinated children reported using only the subset of those with written records with those rates in the overall study population. This may help to strengthen confidence in the accuracy of the overall conclusions

Response to review’s comment: Thank you very much for your concern. The coverage of under vaccinated (34.9%) and fully vaccinated (65.1%) were almost similar among children who had immunization cards and the coverage of under vaccinated (29.4%) and fully vaccinated (59.9%) from either immunization cards or caregivers’ recall. Additionally, caregivers’ recall approach to assess immunization coverage is considered as an accurate and comprehensive approach in the absence of immunization cards if appropriate data collection technique is used. Such method is also recommended by the World Health Organization (WHO) Immunization Coverage Monitoring guideline for assessing childhood vaccination status when immunization cards are not available. However, to make our readers more conscious during the interpretation of our findings, we included the issue of recall bias in the absence of an immunization card at the limitation section.

---

## [Decision Letter · Decision Letter 1]

Vaccination status and its determinants among children aged 12-23 months in Tigray, northern Ethiopia: A zero-inflated Poisson regression analysis

PONE-D-24-50701R1

Dear Dr. Gebretnsae,

We’re pleased to inform you that your manuscript has been judged scientifically suitable for publication and will be formally accepted for publication once it meets all outstanding technical requirements.

Kind regards,

Omar Enzo Santangelo

Academic Editor

PLOS ONE

Additional Editor Comments (optional):

Reviewers' comments:

Reviewer's Responses to Questions

**Comments to the Author**

Reviewer #3: (No Response)

Reviewer #4: All comments have been addressed

Reviewer #5: All comments have been addressed

2. Is the manuscript technically sound, and do the data support the conclusions?

Reviewer #3: Yes

Reviewer #4: Yes

Reviewer #5: Yes

3. Has the statistical analysis been performed appropriately and rigorously?

Reviewer #3: Yes

Reviewer #4: Yes

Reviewer #5: Yes

4. Have the authors made all data underlying the findings in their manuscript fully available?

Reviewer #3: (No Response)

Reviewer #4: Yes

Reviewer #5: Yes

5. Is the manuscript presented in an intelligible fashion and written in standard English?

Reviewer #3: Yes

Reviewer #4: Yes

Reviewer #5: Yes

Reviewer #3: (No Response)

Reviewer #4: All of the editors and reviewer 1 and 2 comments have been addressed by the authors. I have no additional comments.

Reviewer #5: (No Response)

**Do you want your identity to be public for this peer review?** For information about this choice, including consent withdrawal, please see our Privacy Policy

Reviewer #3: No

Reviewer #4: No

Reviewer #5: No

---

## [Editor Report · Acceptance letter]

PONE-D-24-50701R1

PLOS ONE

Dear Dr. Gebretnsae,

I'm pleased to inform you that your manuscript has been deemed suitable for publication in PLOS ONE. Congratulations! Your manuscript is now being handed over to our production team.

Kind regards,

on behalf of

Dr. Omar Enzo Santangelo

Academic Editor

PLOS ONE